# Mutation in the Kinase Domain Alters the VEGFR2 Membrane Dynamics

**DOI:** 10.3390/cells13161346

**Published:** 2024-08-13

**Authors:** Michela Corsini, Cosetta Ravelli, Elisabetta Grillo, Mattia Domenichini, Stefania Mitola

**Affiliations:** 1Department of Molecular and Translational Medicine, University of Brescia, 25123 Brescia, Italy; michela.corsini@unibs.it (M.C.); cosetta.ravelli@unibs.it (C.R.); elisabetta.grillo@unibs.it (E.G.); m.domenichini001@unibs.it (M.D.); 2CN3 “Sviluppo di Terapia Genica e Farmaci con Tecnologia ad RNA”, 25123 Brescia, Italy; 3CIB Consorzio Interuniversitario per le Biotecnologie, 25123 Brescia, Italy

**Keywords:** TK receptors, FLIM/FRET, FRAP, molecular interaction, VEGFR2

## Abstract

Background: Recently, the substitution R1051Q in VEGFR2 has been described as a cancer-associated “gain of function” mutation. VEGFR2^R1051Q^ phosphorylation is ligand-independent and enhances the activation of intracellular pathways and cell growth both in vitro and in vivo. In cancer, this mutation is found in heterozygosity, suggesting that an interaction between VEGFR2^R1051Q^ and VEGFR2^WT^ may occur and could explain, at least in part, how VEGFR2^R1051Q^ acts to promote VEGFR2 signaling. Despite this, the biochemical/biophysical mechanism of the activation of VEGFR2^R1051Q^ remains poorly understood. On these bases, the aim of our study is to address how VEGFR2R1051Q influences the biophysical behavior (dimerization and membrane dynamics) of the co-expressed VEGFR2^WT^. Methods: We employed quantitative FLIM/FRET and FRAP imaging techniques using CHO cells co-transfected with the two forms of VEGFR2 to mimic heterozygosity. Results: Membrane protein biotinylation reveals that VEGFR2^WT^ is more exposed on the cell membrane with respect to VEGFR2^R1051Q^. The imaging analyses show the ability of VEGFR2^WT^ to form heterodimers with VEGFR2^R1051Q^ and this interaction alters its membrane dynamics. Indeed, when the co-expression of VEGFR2^WT^/VEGFR2^R1051Q^ occurs, VEGFR2^WT^ shows reduced lateral motility and a minor pool of mobile fraction. Conclusions: This study demonstrates that active VEGFR2^R1051Q^ can affect the membrane behavior of the VEGFR2^WT^.

## 1. Introduction

Receptor Tyrosine Kinases (RTKs) are single-pass transmembrane proteins with a highly glycosylated extracellular domain and an intracellular kinase domain (TKD), which catalyzes the transfer of the phosphate group from ATP to a tyrosine residue [1]. RTKs are activated by ligand-induced receptor dimerization and/or oligomerization. These induce conformational changes that release the cis-autoinhibition and switch the TKD into its active conformation, eventually leading to transphosphorylation [2]. The phosphorylation of tyrosine is a key post-translational modification that mediates the propagation of extracellular information to intracellular signals by maintaining the active conformation of the kinase itself or becoming docking sites for scaffold proteins. The interactions with membrane lipids (cholesterol, sphingolipids, etc.), influence RTK activation and dynamics [3]. Membrane domains (lipid rafts and caveolae) also modulate the cis-interaction between RTKs, co-receptors, and second messengers. Cis-interactions finely regulate RTK segregation, allowing ligand interaction, dimerization, or interaction with other proteins [4,5,6,7].

The Vascular Endothelial Growth Factor Receptor 2 (VEGFR2) is a 210-230 kDa glycosylated protein expressed by several cells including endothelial and cancer cells. Its expression is regulated under physiological and pathological conditions [8]. Several soluble molecules, including VEGF-A, -C, -D, -F, -E, HIV1-Tat, and gremlin-1, act as receptor ligands [8,9]. VEGFR2 is often dysregulated in cancers, and this is due to an aberrant expression or the acquisition of mutations. Several somatic mutations of VEGFR2 including the substitutions D717V, G800D, G800R, L840F, G843D, R1022Q, R1032Q, R1051Q, D1052N, and S1100F are associated with tumor growth in murine cancer models [10,11]. Importantly, some of them affect the sensitivity to tyrosine kinase inhibitors (TKis), influencing therapeutic choices. Recently, we showed that the expression of VEGFR2^R1051Q^ in cancer cells increases sensitivity to Linifanib. This different receptor behavior can be explained by its higher kinase activity and its ability to activate pro-oncogenic pathways, including the PI3K/Akt/mTOR, and rewire tumor metabolism, thus supporting and enhancing tumor growth in vitro and in vivo [11].

Although the biological effects of VEGFR2^R1051Q^ have been characterized, the biophysical bases of receptor dynamics, which have the potential to modulate receptor function, activation, and response to TKi, remain to be elucidated. Remarkably, the R1051Q substitution occurs in a heterozygous state in cancer patients (cBioPortal https://www.cbioportal.org/ (4 October 2024)). Therefore, in addition to the intrinsic behavior of mutated VEGFR2, it is reasonable to expect that mutated VEGFR2 might influence the behavior of the co-expressed wild-type receptor. On these bases, we analyzed the ability of VEGFR2^R1051Q^ to dimerize with the wild-type receptor altering its membrane dynamics. Using quantitative imaging, the VEGFR2^R1051Q^ membrane dynamics were followed and quantified in live cells. Our data demonstrate that VEGFR2^R1051Q^ heterodimerizes with the wild-type receptor, modifying its membrane dynamics and localization in the absence of ligands, possibly activating cell responses, which may support tumor growth in vivo. 

## 2. Materials and Methods

### 2.1. Cell Cultures

Chinese Hamster Ovary (CHO) cells were grown in Ham’s F-12 (Gibco, ThermoFisher scientific, Carlsband, CA, USA) supplemented with 10% FCS (Gibco) and 1% penicillin/streptomycin. The indicated cells were transiently transfected with the indicated plasmids pBE_hVEGFR2WT [NM_002253.2] (provided by Prof. Kurt Ballmer-Hofer Paul Scherrer Institute, Villigen, Switzerland), pBE_YFP, hVEGFR2^R1051Q^-YFP (obtained as described in [11]), and pBE_hVEGFR2^WT^-mCherry (provided by Dr. Kalina Hristova (Johns Hopkins University, Baltimore, MD, USA) using Polyethyleneimine (PEI, Sigma Aldrich, Milan, Italy). The cells were plated the day before transfection and grown in a complete medium to reach a density of 70–80%. A PEI:DNA complex was prepared at a ratio of 2:1 in a serum-free medium. The day after, a volume of complete medium was added. 

### 2.2. VEGFR2 Immunoprecipitation

To analyze the expression of VEGFR2 onto cell membrane when the indicated membrane proteins were biotinylated, briefly, the confluent cells were incubated for 2 h at 4 °C with biotin-3-sulfo-N-hydroxy-succinimide ester sodium salt (biotin-NHS) (Sigma) dissolved in Hanks’ Balanced Salt Solution (HBSS, ThermoFisher scientific, Carlsband, CA, USA) at 0.5 mg/mL. The cells were lysed in a lysis buffer [50 mM Tris-HCl buffer (pH 7.4) containing 150 mM NaCl, 1% Triton X-100, 1.0 mM Na_3_VO_4_, and protease and phosphatase inhibitors (Sigma)]. Then, 1.0 mg of protein was immunoprecipitated with anti-VEGFR2 (Cell Signaling Technology, Danvers, MA, USA) and separated by SDS–PAGE. The analysis of the biotinylated immunocomplexes was performed using horseradish peroxidase-conjugated streptavidin (HRP-streptavidin, ThermoFisher) and with anti-VEGFR2 antibody (Cell Signaling) in a Western blot. VEGR2 heterodimers were recovered from the lysates of the CHO cells co-transfected with VEGFR2^WT^-YFP/VEGFR2^WT^, VEGFR2^WT^-YFP/VEGFR^R1051Q^, or VEGFR2^R1051Q^ -YFP/VEGFR2^WT^ and incubated for 30 min with 2mM 3,3′-dithiobis(sulfosuccinimidyl propionate) (DTSSP) (ThermoFisher) at 4 °C using anti-GFP antibody (Molecular Probes, ThermoFisher scientific, Carlsband, CA, USA). Immunocomplexes were separated in 6% SDS-PAGE under reducing conditions.

### 2.3. FRAP Analysis

The transfected cells were seeded at 5.0 × 10^5^ cells/mL in µ-slides (ibidi) and analyzed using an LSM880 confocal microscope equipped with an incubation chamber (Carl Zeiss S.P.A, Oberkochen, Germany). The images were recorded with 2% of the intensity of the 514 nm line. YFP was bleached using 50-iteration at 100% intensity of the 514 nm line. Fluorescence recovery in the bleached areas was followed for 7 min (1 image/5 s) and analyzed using the FRAP tool of the Zen black 2.3 software (Carl Zeiss). Data recovered from ROIs were corrected for background and incidental bleaching measured in independent ROIs. Then, the data were fitted using the following formula: I = IE-I1 * exp(−t/T1) where IE is the equilibrium as time goes to infinity and I1 the initial reduction in fluorescence immediately after bleaching.
Diffusion Rate = 1/t_half_   where t_half_ is equal to (ln0.5) * T1

The mobile fraction was determined by comparing the fluorescence in the bleached region after full recovery (F∞) with the fluorescence before bleaching (Fi) and just after bleaching (F_0_). The mobile fraction R was defined as follows: R = (F∞ − F_0_)/(Fi − F_0_) [12].

### 2.4. FLIM/FRET Analysis

To perform fluorescence lifetime imaging microscopy (FLIM), YFP/mCherry was used as FRET pairs, respectively, as the donor and acceptor fluorophores. The fluorescence lifetime of the donor was measured with an LSM880 laser-scanning microscope (Carl Zeiss) equipped with a time-correlated single-photon counting module (PicoQuant, Berlin, Germany). YFP was excited at 860 nm using a picosecond-pulsed Chameleon Vision II laser at an 80 MHz repetition rate (Coherent Inc, Santa Clara, CA, USA). The time-correlated single-photon counting decay curves were fitted using the SymphoTime 64 version 2.2 software (PicoQuant, Germany). The average lifetime of YFP was measured in the membrane region of interest (ROI), and the reference lifetime was measured in the CHO cells expressing ECD-VEGFR2-YFP. The FRET efficiency (E) was calculated using the following equation where τD and τDA are the lifetime values of the donor fluorophore in the absence and presence of the acceptor, respectively.
E = 1 − (τDA/τD)

The radius (R) was calculated using the following equation where R_0_ is the Förster radius:R = R0 − [1/E − 1]^1/6^

### 2.5. Statistical Analyses

The data were analyzed using Prism10 (GraphPad Software version 10.2.3). Student’s *t*-test for unpaired data (2-tailed) was used to test the probability of significant differences between the two groups of samples. The differences were considered significant when *p* *, *p* < 0.05; **, *p* < 0.01; ***, *p* < 0.001; and ****, *p* < 0.0001. For more than two groups of samples, the data were statistically analyzed with one-way ANOVA, and individual group comparisons were evaluated by the Bonferroni multiple comparison test. The error bars in the graphs represent the standard error of the mean (SEM).

## 3. Results

### 3.1. Mutation R1051Q Modulates the VEGFR2 Turnovers on Cell Membrane

To clarify the mechanism that supports and enhances tumor growth in the cells expressing the constitutive active VEGFR2^R1051Q^, in a first set of experiments, we assessed if the presence of the substitution R1051Q in VEGFR2 modulates its behavior on the membrane. The amount of receptor on the cell membrane was evaluated on VEGFR2 immunocomplexes recovered from biotinylated VEGFR2^WT^ and VEGFR2^R1051Q^-expressing CHO cells. Blotting with HRP-streptavidin showed that VEGFR2^R1051Q^ was less exposed in the membrane compared to the VEGFR2^WT^ (Figure 1). Similar results were obtained in human breast cancer MCF7 cells. Of note, VEGF stimulation induced a rapid internalization of VEGFR2^WT^ while promoting the exposure of VEGFR2^R1051Q^ as demonstrated by the different levels of biotinylated VEGFR2. (Appendix A). These data, together with their constitutive phosphorylation [11,12], suggest a more rapid membrane turnover of VEGFR2^R1051Q^ with a higher receptor recycled in the membrane.

### 3.2. VEGFR2^R1051Q^ Heterodimerizes with Wild-Type Receptor

Since in cancer, VEGFR2^R1051Q^ is expressed in a heterozygous state, we evaluated whether its pro-oncogenic ability may be, at least in part, mediated by the heterodimerization with co-expressed wild-type receptor. To visualize and quantify the possible heterodimerization between the wild-type and mutated receptors in live cells, we performed Forster Resonance Energy Transfer (FRET) analysis based on fluorescence lifetime imaging microscopy (FLIM) on the membrane of the CHO cells [13,14]. The lifetime is an intrinsic property of each fluorophore and can be influenced by its molecular environment. In FRET experiments, interaction with the acceptor leads to a reduction in donor fluorescence lifetime. The evaluation of this shortening allows FRET efficiency calculation.

FLIM/FRET analysis overcomes the main bias of intensity-based FRET calculation that relies on the dependency of FRET estimation on the expression of both the donor (YFP) and the acceptor (mCherry) fluorescence and on their quantum yield requiring reference and calibration.

Importantly, in all the experiments, we used YFP as the donor fluorophore and we measured only its lifetime decay variation. In our experimental condition, the YFP alone showed a lifetime of 2.3 ns. Although YFP displays a single exponential fluorescence lifetime decay when expressed in the absence of other fluorescent molecules [15,16], we analyzed the FLIM/FRET experiments using a double exponential lifetime decay to highlight the receptor pool possibly dimerizing with the mCherry-tagged receptor. To this, CHO cells were alternatively co-transfected with VEGFR2^WT^-YFP or VEGFR2^R1051Q^-YFP with VEGFR2^WT^-mCherry. When dimerization occurs, the different pools of YFP-tagged receptors can be identified, each with a specific lifetime. Indeed, the interaction between two tagged receptors leads to the proximity of the donor and the acceptor with a consequent transfer of energy and the reduction in YFP lifetime value. The donor mean lifetime (YFP tagged VEGFR2) was 1.524 ns in VEGFR2^R1051Q^-YFP/VEGFR2^WT^-mCherry, which is significantly lower than that in VEGFR2^WT^-YFP/VEGFR2^WT^-mCherry (Figure 2A,B). By imaging analysis, we calculated the distance between the two molecules (Figure 2C) and FRET efficiency (Figure 2D–F). As expected in the VEGFR2^R1051Q^-YFP/VEGFR2^WT^-mCherry-expressing cells, the FRET efficiency (35.86%) was higher than in the cells co-expressing VEGFR2^WT^ with both YFP- and mCherry-tagged receptors. The color code imaging of FRET efficiency suggested an increased probability of dimerization between VEGFR2^R1051Q^ and VEGFR2^WT^ with respect to VEGFR2^WT^-VEGFR2^WT^ which reached 24.48% (Figure 2D,E). In keeping with these data, the occurrence in FRET efficiency distribution (Figure 2F) showed a significant shift in the VEGFR2^R1051Q^-YFP/VEGFR2^WT^-mCherry-expressing cells, suggesting potential differences in receptor interactions or conformational states between the two receptors. Finally, the VEGFR2^WT^/VEGFR2^R1051Q^ heterodimer was immunoprecipitated using anti-GFP from CHO cells co-expressing VEGFR2^WT^-YFP/VEGFR2^WT^, VEGFR2^WT^-YFP/VEGFR2^R1051Q^, or VEGFR2^R1051Q^-YFP/VEGFR2^WT^. To this, the cells were treated with 2 mM of DTSSP and lysate. The VEGF-treated VEGFR2^WT^-YFP/VEGFR2^WT^ CHO cells were used as positive control. Figure 2G shows the presence of untagged receptors in the GFP-immunocomplexes. 

### 3.3. VEGFR2^R1051Q^ Affects the Lateral Mobility of Wild-Type Receptor

The presence of hetero complexes in co-expressing cells induced us to investigate the reciprocal influences of wild-type and mutated receptors on their membrane dynamics. To this VEGFR2^R1051Q^-YFP or VEGFR2^WT^-YFP were transfected in CHO cells, which express undetectable levels of endogenous receptor, alone or with untagged VEGFR2^WT^ to mimic the patient heterozygous state. The lateral diffusion of VEGFR2^R1051Q^-YFP was characterized by Fluorescence Recovery After Photobleaching (FRAP) analysis. To this, the fluorescence intensity of different membrane areas was measured before and after photobleaching every 5′′ for 7 min. As expected, the VEGFR2^R1051Q^ variant showed a lower diffusion rate compared to VEGFR2^WT^ (0.01715 vs. 0.02665 1/s) (Figure 3A,B). Then, to assess whether the mutated receptor alters the dynamic of VEGFR2^WT^, we compared the dynamics of VEGFR2^WT^-YFP in the absence or in the presence of untagged VEGFR2^R1051Q^. In the above FRAP analysis, only YFP-tagged receptors were followed, allowing us to identify the possible influences of untagged receptors on tagged ones (Figure 3C). When we compared the membrane behavior of the single YFP-tagged receptors, we noticed that the VEGFR2^R1051Q^-YFP mobile fraction had a higher lateral diffusivity (Figure 3A) and that the amount of mobile VEGFR2^R1051Q^-YFP was lower with respect to that of wild-type receptor (Figure 3B) as described by the half time of fluorescence recovery assay. This is due to the diffusion rate, which well describes a receptor in its activated conformation. While the co-expression of the wild-type receptor did not significantly affect the dynamics of VEGFR2^R1051Q^-YFP (Figure 3A,B), a significant increase in the percentage of the mobile fraction of VEGFR2^WT^-YFP parallel with a reduction in its lateral diffusivity was detected when co-expressed with VEGFR2^R1051Q^.(Figure 3A–D) These altered membrane dynamics, together with the evidence of VEGFR2^R1051Q^/VEGFR2^WT^ heterodimerization, suggested the ability of active VEGFR2^R1051Q^ to influence the membrane mobility and function of the wild-type receptor in the absence of any ligands.

## 4. Discussion

The VEGF/VEGFR2 system regulates cancer cell proliferation, motility, metabolism, and stemness. Thus, in recent years, the VEGF/VEGFR2 system has become a therapeutic target in cancer treatment [17]. The use of antibodies against VEGF or VEGFR2, as well as the VEGFR2-targeted TKi (i.e., pazopanib, cediranib, etc.) exerts a dual effect on cancer. They reduce tumor growth directly by inhibiting parenchymal cells and indirectly altering the angiogenic process. The increased NGS data have highlighted a high degree of RTK alteration including VEGFR2 in human tumors. A total of 21 different mutations in the KD of VEGFR2 [10] were identified but the pro-oncogenic role is characterized only for a few of them. Recently, we described the metabolic and pro-oncogenic effects of the substitution R1051Q in the activation loop of the KD of VEGFR2. Here, we deepen the biophysical behavior/membrane turnover, lateral mobility, and dimerization of VEGFR2^R1051^ and how this might affect the wild-type ones. We found that the mutant form of VEGFR2 is endowed with intrinsically altered membrane turnover and dynamics. In addition, VEGFR2^R1051Q^ dimerizes and controls the membrane behavior of co-expressed VEGFR2^WT^. VEGFR2 has been described to exist in a monomeric form in the absence of a ligand and follows the canonical model of ligand-induced dimerization and activation [18]. However, it has also been demonstrated that full-length VEGFR2 forms dimers in the absence of ligands [1]. Previous evidence highlighted the ability of several RTKs including EGFR, FGFRs, IR, TRK [19,20], and PDGFRs [21] to dimerize in the absence of ligands. This dimerization allows the receptors to acquire a ready-to-active conformation to be rapidly activated following interaction with ligands. We found that the mutant form of VEGFR2 can dimerize with VEGFR2^WT^ in the absence of ligands, and slows down its lateral mobility on the cell membrane, possibly affecting its biological (signaling) behavior. Our data point to the possibility that mutated VEGFR2 may spatially relocate the heterodimeric complex towards membrane micro-domains where active receptors often localize [22,23]. However, this remains to be elucidated.

Evidence point out that RTK dimerization is required, but is not sufficient, for kinase activation. RTKs require cross-phosphorylation for a complete activation. This process is allowed by the specific orientation of the KD in the dimerization. Usually, ligand interactions optimally orient the KDs as a result of structural rearrangements that propagate along the length of the RTK. RTK transmembrane domain also rotates upon ligand binding, leading to periodic oscillations in the KD which increase its phosphorylation [1,22]. Thus, we can speculate that the substitution R1051Q might stabilize the activable receptor conformation supporting dimerization and signaling even in the absence of ligands. The VEGFR2^R1051Q^ might alter the efficiency of the receptor system by making it more sensitive to changes in the duration of specific interactions. Of course, further studies will be necessary to verify this hypothesis. This suggests that this mutation may have a substantial impact on the overall behavior of the VEGFR2 system, particularly in the context of cancer. 

## 5. Conclusions

In summary, this study provides valuable insights into the molecular dynamics of VEGFR2 and its mutant form VEGFR2^R1051Q^, identified in cancer patients. The spatial regulation and the dynamics of RTKs in the membrane also influence the fate, position, or polarity of cells in tissues. In epithelial cells as well as in endothelial cells, receptor relocation maintains cell polarization [23]. Since the spatial distribution of RTKs is so finely regulated, the RTK dynamics represent a field of study that needs to be deepened especially with regard to the tumor field.

## Figures and Tables

**Figure 1 cells-13-01346-f001:**
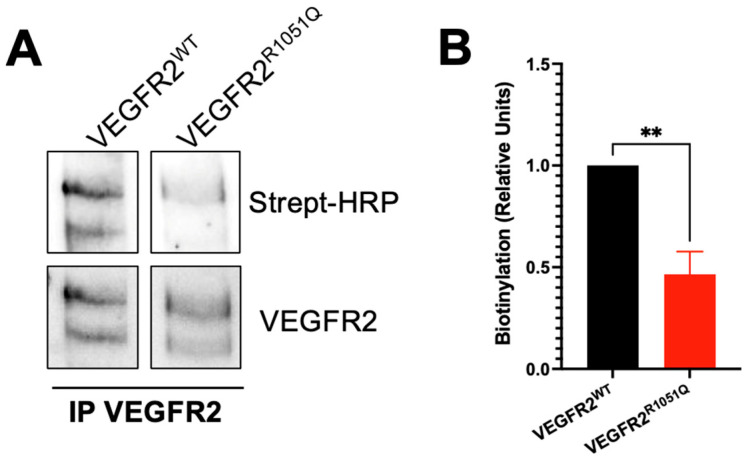
*VEGFR2^R1051Q^ is less exposed on the cell membrane.* (**A**) VEGFR2^WT^ and VEGFR2^R1051Q^ were immunoprecipitated from biotinylated CHO cells and blotted for Streptavidin-HRP. VEGFR2 was used as a normalizer. (**B**) The Western blot quantification of three independent experiments (**, *p* < 0.01).

**Figure 2 cells-13-01346-f002:**
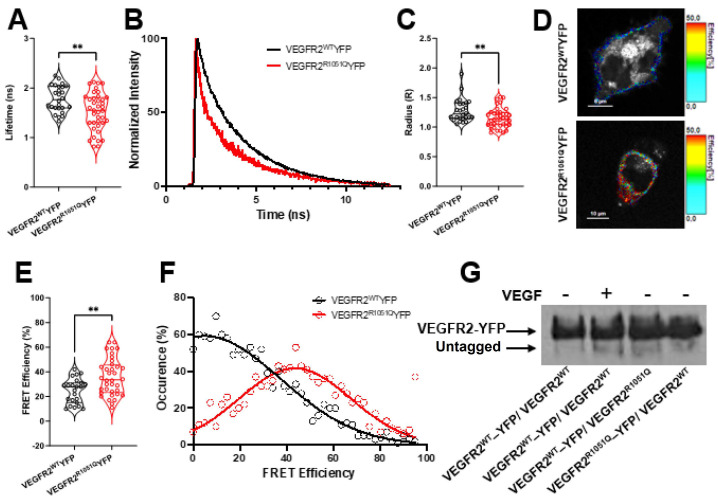
*VEGFR2^WT^ dimerizes with VEGFR2^R1051Q^*. The FLIM/FRET analysis of the interaction between VEGFR2^R1051Q^-YFP or VEGFR2^WT^-YFP with VEGFR2^WT^-mCherry in the CHO cells (n = 20–25 cell measurements, **, *p* < 0.01). (**A**) The lifetime of VEGFR2^WT^-YFP or VEGFR2^R1051Q^-YFP in the presence of VEGFR2WT-mCherry (**, *p* < 0.01). (**B**) The representative decay curves of VEGFR2^R1051Q^-YFP or VEGFR2^WT^-YFP in the presence of VEGFR2^WT^-mCherry. (**C**) The distance of VEGFR2^WT^-YFP or VEGFR2^R1051Q^-YFP and VEGFR2^WT^-mCherry. (**D**) Representative color-coded FRET efficiency in VEGFR2^WT^-YFP/mCherry-VEGFR2^WT^ and VEGFR2^R1051Q^-YFP/VEGFR2^WT^-mCherry. (**E**) FRET efficiency (%) distribution. (**F**) Occurrence distribution in a representative cell membrane. (n = 20–25 cell measurements, **, *p* < 0.01). (**G**) WB anti-VEGFR2 of immunocomplexes recovered using anti GFP from lysed VEGFR2^WT^-YFP/VEGFR2^WT^, VEGFR2^WT^-YFP/VEGFR2^R1051Q^, or VEGFR2^R1051Q^-YFP/VEGFR2^WT^ CHO cells upon treatment with 2mM of DTSSP crosslinker.

**Figure 3 cells-13-01346-f003:**
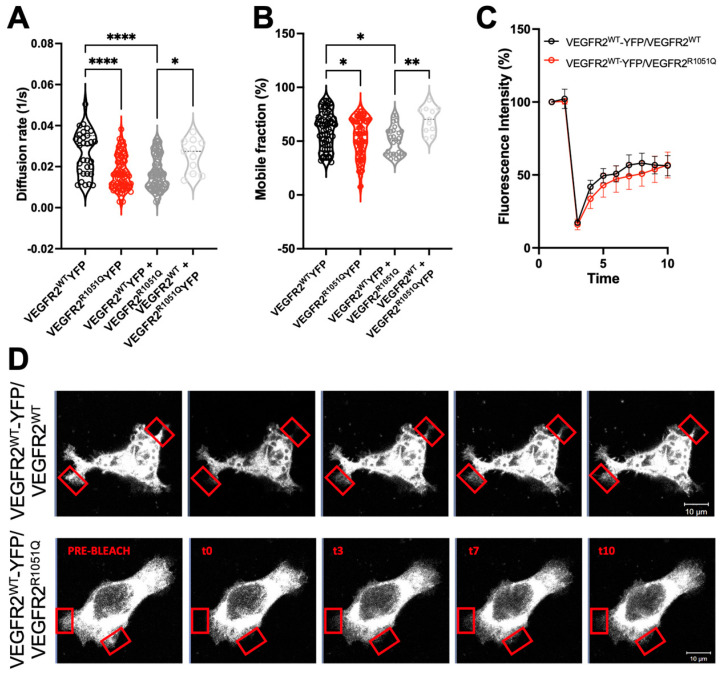
*VEGFR2^R1051Q^ alters the membrane dynamics of the wild-type receptor.* FRAP analysis was performed on the plasma membrane of CHO transfected with VEGFR2^WT^-YFP, VEGFR2^R1051Q^-YFP in the absence or in the presence of untagged VEGFR2^WT^, or VEGFR2^R1051Q^. (**A**) Diffusion coefficient (1/s); (**B**) mobile fraction (%); (**C**) Representative fluorescence recovery curves; (**D**) representative images acquired for 7 min every 5 s. The bleached areas are indicated by a red square (white bar: 10 μm). The data are representative of n = 20–25 cell measurements. One-way ANOVA was applied associated with the Bonferroni multiple comparison test (*, *p* < 0.05; **, *p* < 0.01; and ****, *p* < 0.0001).

## Data Availability

All data are included in the present paper.

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
