# Peer review of "Mutation in the Kinase Domain Alters the VEGFR2 Membrane Dynamics"

_cells, 2024, doi:10.3390/cells13161346_

Round 1

Reviewer 1 Report

Comments and Suggestions for Authors

The authors emphasize the significance of RTK membrane dynamics and spatial regulation in cancer. By examining the VEGFR2R1051Q mutation using FLIM/FRET and FRAP techniques in a heterozygous cell model, the authors found that this mutation affects receptor localization and heterodimerization even without ligand interaction, suggesting its inherent oncogenic potential. However, in this reviewer’s opinion, improvements need to be made before considering publication.

Main points:

1.       The author should describe their mathematical models about how they calculate the distance between protein molecules (Figure 2C), how they calculate diffusion rate and mobile fraction (Figure 3A and 3B).

2.       It is interesting that the mutant VEGFRR1051QYFP showed lower diffusion rate by itself (Figure 3A and 3B). The authors should provide explanations about this phenotype. In addition, the data from co-expression of WT VEGFR with VEGFRR1051QYFP should not be omitted (Line 203) as it is conducted in the same set of experiment.

3.       Given that there is decreased level of mutant VEDFR proteins on the membrane, could the lower diffusion rate in Figure 3A is due to the lower protein abundance? Again, detailed mathematical model is necessary to help audience to appreciate their claims.

4.       The authors claimed that the decreased expression level and diffusion rate are related to the oncogenic nature of this mutation. They should provide evidence of the downstream activity to prove its tumor causing effects.

Minor points:

1.       Line 124, turnover normally means the rate of synthesize and degradation. However, results presented in Figure 1 do not support this claim. The authors should choose other terms.

2.       Line 142-144 is repeated in the next paragraph.

3.       Line 123, data from MCF7 cell line should not be omitted because these receptors may not behave correctly in CHO cell line.

4.       Statistically, the author should perform ANOVA test when more than two conditions exist (Figure 3). Detailed statistical values should be provided in each figure legend, including t value, degree of freedom and p value.

Comments on the Quality of English Language

I suggest the authors check their typos and smoothen their language to aid the understanding of audience.

Author Response

Reviewer #1

  1. The author should describe their mathematical models about how they calculate the distance between protein molecules (Figure 2C), how they calculate diffusion rate and mobile fraction (Figure 3A and 3B).

We thank the reviewer for the suggestion. We implemented the Materials and Methods with details of FRAP analysis (see lines 91-97) with: 

“ Data recovered from ROIs were corrected for background and incidental bleaching measured in independent ROIs. Then data were fitted using the follow formula:  I=IE-I1*exp(-t/T1).

 Diffusion Rate=1/thalf     were thalf is equal to (ln0.5)*T1.

The mobile fraction was determined by comparing the fluorescence in the bleached region after full recovery (F∞) with the fluorescence before bleaching (Fi) and just after bleaching (F0). The mobile fraction R was defined as : R= (F∞- F0)/( Fi- F0)”  . 

Also, we inserted the formula for Radius calculation (see lines 111-112):

“Radius (R) was calculated using the following equation where R0 is the Förster radius:  R0−[1/E -1]â…™

Also we added a new reference:  E.A. Reits, J.J. Neefjes, From fixed to FRAP: measuring protein mobility and activity in living cells, Nature cell biology, 3 (2001) E145-147.

  1. It is interesting that the mutant VEGFRR1051QYFP showed lower diffusion rate by itself (Figure 3A and 3B). The authors should provide explanations about this phenotype. In addition, the data from co-expression of WT VEGFR with VEGFRR1051QYFP should not be omitted (Line 203) as it is conducted in the same set of experiment.

      According to the reviewer’s request, we added the FRAP experiment coexpressing VEGFR2-WT with VEGFR2 R1051Q-YFP in the revised version (see Fig 3A-B). In the previous version, we omitted it because the paper was focused only on the effects of the constitutively active receptor. Indeed VEGFR2 WT is unphosphorylated and inactive in VEGF-untreated cells thus as expected it did not affect VEGFR2 R1051Q behaviors.       

  1. Given that there is decreased level of mutant VEDFR proteins on the membrane, could the lower diffusion rate in Figure 3A is due to the lower protein abundance? Again, detailed mathematical model is necessary to help audience to appreciate their claims.

Lateral diffusion of the receptor can be affected by several factors including temperature, membrane lipid content, and others. To address this, we strictly maintained experimental conditions of temperature, CO2, and cell confluence. Also we selected cells with comparable fluorescence intensity for FRAP analysis to exclude the mere effects of the expression levels. 

  1. The authors claimed that the decreased expression level and diffusion rate are related to the oncogenic nature of this mutation. They should provide evidence of the downstream activity to prove its tumor causing effects

In our previous papers (Grillo et al. doi: 10.1016/j.canlet.2021.03.007., Domenichini et al. doi: 10.1016/j.bbagen.2023.130470) we described the pro-oncogenic potential of VEGFR2 R1051Q in vitro and in vivo. As published (Grillo et al. Cancer Letters 2020 - Fig.5) the co-expression modifies the intracellular signaling of melanoma cancer cells. In particular, differences in the activation of the PI3K/Akt/mTOR pathway were observed. In accordance, the AKT/mTOR pathway inhibitors AZD8055, rapamycin, and everolimus inhibit the proliferation of Sk-Mel-31-VEGFR2R1051Q with higher efficacy if compared with Sk-Mel-31-VEGFR2WT as well as the mutation alters the sensitivity of Linifanib. Again the expression of VEGFR2 R1051Q alters the metabolic profile of melanoma cells which results in more addiction to glutamine. The aim of this paper was to characterize how and if VEGFR2 R1051Q may alter the VEGFR2 wt behaviors. 

Minor points:

  1.       Line 124, turnover normally means the rate of synthesize and degradation. However, results presented in Figure 1 do not support this claim. The authors should choose other terms.

We modified the sentence “This data together with its constitutive phosphorylation (REF) suggests a higher turnover of VEGFR2R1051Q respect to the VEGFR2WT.” as “This data together with its constitutive phosphorylation (REF) suggests a rapid internalization of VEGFR2R1051Q with respect to the VEGFR2WT. Also we added a supplementary Fig 1 to highlight the VEGFR2 membrane  turnover in untreated and treated cells 

  1. Line 142-144 is repeated in the next paragraph.

We apologize for this oversight. We have made the necessary changes to the text as requested.

  1.       Line 123, data from MCF7 cell line should not be omitted because these receptors may not behave correctly in CHO cell line.

We thank the reviewer for the suggestion. The similar data obtained with MCF7 cell lines, which are not expressing VEGFR2 as CHO cells, will be inserted in Supplementary Figure 1 and commented in lines 135-140.

  1.       Statistically, the author should perform ANOVA test when more than two conditions exist (Figure 3). Detailed statistical values should be provided in each figure legend, including t value, degree of freedom and p value.

All statistical analysis tests were described in the Material and Methods section (lines 113-119). In the revised version we added the statistical details also in the figure legends (see legend Fig 3).

Reviewer 2 Report

Comments and Suggestions for Authors

In the article, the authors investigated the impact of the VEGFR2R1051Q mutation on receptor tyrosine kinase (RTK) dynamics. RTKs play a crucial role in cancer by activating upon ligand interaction, leading to dimerization and phosphorylation. The study uses imaging techniques in CHO cells co-transfected with both wild-type and mutated VEGFR2 to mimic the heterozygous state found in tumors. Results indicate that the mutation affects membrane exposure and localization of the receptor, causing it to form heterodimers with the wild-type receptor even without ligands. This emphasizes altered membrane dynamics due to the mutation, highlighting the importance of understanding such dynamics in cancer. The findings offer potential insights for targeted therapies in VEGFR2-related tumors.

Comments

1.     The authors claim that wild-type and mutated VEGFR2 form heterodimerization. It would be helpful if the authors explained more why wild-type and mutant VEGFR2 form heterodimerization.

2.     What is the rationale to use only VEGFR2 1052Q mutant to validate heterodimeration instead of other mutants such as D717V, G800D, G800R, L840F, G843D, R1022Q, R1032Q, R1051Q, D1052N, and S1100F.

3.     In addition to IPs, authors have the option to perform Native-PAGE to confirm the presence of the heterodimer complex. However, MD simulations would be more suitable for a deeper understanding of the mechanistic insights behind heterodimerization and membrane interactions.

4.     In addition, the authors could consider showing VEGFR2 wild-type or mutant phosphorylation on western blot, phosphotag gels, or P32 assays.

5.     Line 56, what is TKis? tyrosine kinase inhibitor?

Comments on the Quality of English Language

The quality of the English looks good. 

Author Response

  1. The authors claim that wild-type and mutated VEGFR2 form heterodimerization. It would be helpful if the authors explained more why wild-type and mutant VEGFR2 form heterodimerization.

RTKs play crucial roles in cellular processes. Due to the heterozygous state of somatic mutation of VEGFR2, cancer cells express both wild-type and mutant receptors. Heterodimerization of RTKs is a mechanism that enhances the diversity and specificity of signaling responses. As published (Grillo et al. Cancer Letters 2020 - Fig.5) the co-expression modifies the intracellular signaling of melanoma cancer cells.

  1. What is the rationale to use only VEGFR2 1052Q mutant to validate heterodimeration instead of other mutants such as D717V, G800D, G800R, L840F, G843D, R1022Q, R1032Q, R1051Q, D1052N, and S1100F.

Although we cannot exclude that other mutated receptors may heterodimerize with the wild type, a pan-cancer analysis identified 10 hot spot mutations in the kinase domain of several receptors, some of which are expressed in a heterozygous state. In particular, receptors with the substitution in position 256, correspond to R1051Q in VEGFR2. Hence the idea of a possible heterodimerization. Furthermore, preliminary molecular modeling on VEGFR2 R1051Q highlighted conformational changes in the activation loop of the kinase domain similar to that identified in VEGF-stimulated receptors. Other mutations are currently under study.

In addition to IPs, authors have the option to perform Native-PAGE to confirm the presence of the heterodimer complex. However, MD simulations would be more suitable for a deeper understanding of the mechanistic insights behind heterodimerization and membrane interactions.

We thank the reviewer for the suggestion. In the alternative of Native-PAGE or MD simulations, to show the heterodimers of VEGFR2WT/VEGFR2R1051Q we performed a crosslinking experiment on CHO co-transfected with VEGFR2WT-YFP/VEGFR2WT, VEGFR2WT-YFP/VEGFR2R1051Q or VEGFR2R1051Q-YFP/VEGFR2WT . Before cell lysate membrane proteins were crosslinked with DTSSP. DTSSP does not pass the membrane and contains a S-S bound which can be destroyed under reducing conditions. Thus tagged receptors were immunoprecipitated from CHO lysates using anti-GFP (of note anti-GFP recognizes also YFP) . Finally, GFP-immunocomplexes were treated with a reducing sample buffer, separated onto SDS-PAGE gel and assessed with anti-VEGFR2. Under non-reducing conditions the untagged receptors precipitated with the GFP-immunocomplex. Anti-VEGFR2 revealed 2 different bands, the higher is the VEGFR2-YFP while the lower is the untagged ones. Blot was included in Fig. 2G, and the entire acquisition is available in Supplementary Fig.

In addition, the authors could consider showing VEGFR2 wild-type or mutant phosphorylation on western blot, phosphotag gels, or P32 assays.

 Line 56, what is TKis? tyrosine kinase inhibitor?

Yes, TKi are tyrosine kinase inhibitors, as written in the paragraph on the line (48-49). The addition of the 's' is a typographical error.

Reviewer 3 Report

Comments and Suggestions for Authors

Using an in vitro CHO model, the authors of this paper reported that VEGFR2 R1051Q mutant exhibits different biophysical properties than the wild-type protein. In addition, VEGFR2 R1051Q forms heterodimer with wild-type protein and thereby affecting the membrane dynamics and localization of wild-type protein in the absence of ligands. 

Major concerns: 1. There are no references that establish use of FLIM-FRET to measure heterodimerization; 2. The quality of western blots in Figure 1 is poor. The original blots images should be provided to increase credibility. 3. In Figure 2D, there are lots of dots with high FRET efficiency in the cytoplasmic region of VEGFR2R1051Q-YFP panel, what are the significance of those hot spots? Are those cytoplasmic dots excluded in the analysis in 2E? 4. There are two places where authors mentioned some results without showing data (“data not shown”). They should either show the data or exclude the results.

 Minor issues: 1. Duplication of sentences 142-145 with 146-148; 2. There are several typos such as at line 212 (“dynamicsof wilfd type receptros”) and line 216 (“rappresentative”).

Author Response

Major concerns: 

  1. There are no references that establish use of FLIM-FRET to measure heterodimerization.

Thank you for the suggestion. We inserted two references in lines 151, in the first the authors described the HGF-induced dimerization of tyrosine kinase receptor MET [doi.org/10.1016/j.bbamcr.2016.04.01] while in the second [DOI: 10.1016/j.jmb.2020.06.009] the authors described the dimerization of GPR.

  1. The quality of western blots in Figure 1 is poor. The original blots images should be provided to increase credibility. 

The entire western blots are included in Supplementary Fig.

  1. In Figure 2D, there are lots of dots with high FRET efficiency in the cytoplasmic region of VEGFR2R1051Q-YFP panel, what are the significance of those hot spots? Are those cytoplasmic dots excluded in the analysis in 2E? 

For the calculation of FRET efficiency a membrane ROI was selected, while we inserted the entire picture of FRET efficiency. 

  1. There are two places where authors mentioned some results without showing data (“data not shown”). They should either show the data or exclude the results. 

Thank you for the suggestion. As also highlighted by the other reviewer, data not shown were included in the manuscript as Supplementary Figure 1.

 Minor issues: 1. Duplication of sentences 142-145 with 146-148; 2. There are several typos such as at line 212 (“dynamicsof wilfd type receptros”) and line 216 (“rappresentative”).

We have carefully reviewed the text to correct any typos or typographical errors.

Reviewer 4 Report

Comments and Suggestions for Authors

The manuscript presents an analysis comparing the behavior of point mutants within the kinase domain of VEGFR2 on the cell membrane with that of the wild-type VEGFR2. The study shows a certain degree of completeness in the methods designed, results presented, and interpretations made. While the manuscript demonstrates the behavior of point mutants within the kinase domain of VEGFR2 on the cell membrane, it lacks an examination of the physiological impact on cancer cell characteristics. Specifically, there is no investigation into the differences in sensitivity to various drugs targeting VEGFR2, nor is there a comparison of oncogenic potential using cancer models, such as tumor formation in animal models. Given the preliminary nature of the findings, the manuscript does not meet the standards required for acceptance in the journal "Cells" at this stage. Further comprehensive studies are necessary to explore the physiological relevance and implications of the point mutations in VEGFR2 within a broader oncological context.

Author Response

We recently demonstrated the pro-oncogenic activity of the VEGFR2R1051Q and its effects on cancer cell metabolism (https://doi.org/10.1016/j.canlet.2020.09.027;  https://doi.org/10.1016/j.canlet.2021.03.007 ) using both in vitro and in vivo models. We identified this mutation using a pan-cancer analysis. This analysis highlighted a mutational hot spot in position 256 of the kinase domain of several receptors. Again in (doi: 10.1016/j.bbagen.2023.130470) we tested the effects of tyrosine kinase inhibitors in in vitro and in vivo assays suggesting for example that the linifanib shows a lower IC 50 for VEGFR2R1051Q with respect to wild type. In this paper, we deepen the molecular mechanism that supports this oncogenic potential.

Round 2

Reviewer 3 Report

Comments and Suggestions for Authors

Authors have addressed majority of concerns.

Author Response

Review :Authors have addressed majority of concerns.

We would like to thank the referee who helped us to improve the quality of paper

Reviewer 4 Report

Comments and Suggestions for Authors

I have reviewed the response from the paper's authors.  They merely explained their previous research history on the function of mutant proteins.  This does not actively address the essence of my comment, which is: "For acceptance of the paper in Cells, its content being limited to the intracellular reconstitution of mutant proteins without delving into their physiological significance leads to a negative conclusion."  Therefore, I cannot consider this as a valid response that warrants a re-review.

Author Response

We regret the comments of this reviewer, we believe that he/she did not understand the purpose of the work. Our work only aims at the  explanation of how a mutated receptor can also alter the response of wild type ones. The choice of the R1051 mutation of VEGFR2 was dictated by the previous published results of  the same laboratory in which only the  biological activities were characterized expressing VEGFR2R1051Q in cells that expressed or not the endogenous wild type receptor. This was to mimic the condition of heterozygosity identified in cancer patients. While, here we analyzed the ability of VEGFR2R1051Q  to dimerize with VEGFR2WT affecting its membrane dynamics